# Preliminary Data about Habitat Use of Subadult and Adult White Sharks (*Carcharodon carcharias*) in Eastern Australian Waters

**DOI:** 10.3390/biology11101443

**Published:** 2022-10-01

**Authors:** Jessica L. Coxon, Paul A. Butcher, Julia L. Y. Spaet, Justin R. Rizzari

**Affiliations:** 1School of Life and Environmental Sciences, Deakin University, Waurn Ponds, VIC 3216, Australia; 2New South Wales Department of Primary Industries, Fisheries, National Marine Science Centre, Southern Cross University, Coffs Harbour, NSW 2450, Australia; 3Evolutionary Ecology Group, Department of Zoology, University of Cambridge, Cambridge CB2 3EJ, UK

**Keywords:** archival tag, diel behaviour, horizontal movement, migration, Pacific Ocean, satellite telemetry, shark management, vertical migration

## Abstract

**Simple Summary:**

In eastern Australia, the movement of subadult and adult white sharks is poorly understood as very few animals are encountered. To investigate how deep they dive, where they travel and water temperatures they live in, we tagged three white sharks (340–388 cm total length) between May 2021 and January 2022. All sharks moved away from the coast after release and preferred offshore habitats. The three sharks mainly stayed within 50 m of the surface, in water temperatures between 14–19 °C and dived deeper during the day. They all moved north–south along the coastline and spent little time in the same location but there was evidence that they stayed in some locations longer than others. Very little information is available for adult white sharks in eastern Australia and studies like this provide key information about their preferences. As they travel long distances it is important for multijurisdictional management to effectively mitigate human-shark interactions whilst supporting conservation efforts.

**Abstract:**

In eastern Australia, white sharks (*Carcharodon carcharias*) are targeted in shark control programs, yet the movement of subadults and adults of the eastern Australasian population is poorly understood. To investigate horizontal and vertical movement and habitat use in this region, MiniPAT pop-up satellite archival tags were deployed on three larger white sharks (340–388 cm total length) between May 2021 and January 2022. All sharks moved away from the coast after release and displayed a preference for offshore habitats. The upper < 50 m of the water column and temperatures between 14–19 °C were favoured, with a diel pattern of vertical habitat use evident as deeper depths were occupied during the day and shallower depths at night. Horizontal movement consisted of north–south seasonality interspersed with periods of residency. Very little information is available for adult white sharks in eastern Australia and studies like this provide key baseline information for their life history. Importantly, the latitudinal range achieved by white sharks illuminate the necessity for multijurisdictional management to effectively mitigate human-shark interactions whilst supporting conservation efforts of the species.

## 1. Introduction

White sharks (*Carcharodon carcharias*, Linnaeus 1758) are a large, circumglobally distributed, apex predator that play an important regulatory role in many marine habitats from the continental shelf to pelagic seas, and in surface waters down to depths of 1280 m [1,2,3,4,5]. In Australia, white sharks are broadly distributed in southern waters, primarily ranging from North West Cape, Western Australia to central Queensland [6,7,8]. Previous studies have demonstrated restricted east–west movement across Bass Strait in southern Australia, leading to a two-population model for the Australasian region, which has been supported using acoustic telemetry, genetic analysis, and genetic assessment of kinship [9,10,11,12]. However, there is connectivity between populations [13]. The western population, referred to as the southern-western Australasian population (SWP), primarily extends from Wilsons Promontory, Victoria to North West Cape, Western Australia, with connectivity between South Africa [14]. The eastern population, hereafter referred to as the eastern Australasian population (EAP), extends from Wilsons Promontory, Victoria to central Queensland and waters surrounding New Zealand [11,15,16]. 

Understanding white shark movement and distribution is paramount in illuminating key habitats and informing management actions [17] and has thus, been identified as one of the top 10 global research priorities for the species [18]. Movement of the EAP has been well documented [12,15,16]. Increased sample sizes and more comprehensive analyses revealed movement of the EAP to extend along the entire length of the eastern seaboard of Australia, expanding the distribution past central Queensland [13]. Seasonal movement is evident with peak abundances primarily documented in New South Wales and Queensland during the months May to November [1,13,16,19] supported by peak catch rates in shark management programs [20]. Tracking the seasonal movement of juvenile white sharks (<300 cm Total Length—TL) led to the discovery of two resident nursery areas at Corner Inlet, Victoria and Port Stephens, New South Wales in south-eastern Australia, and identified a north–south corridor utilised during migration between these two locations [1]. This evidence highlights the propensity for white sharks to occupy coastal waters, where human-shark interactions, such as commercial fishing and shark control programs, have led to negative impacts on population densities. 

In 2015, following an increased number of unprovoked shark bites and a fatality, the New South Wales Government introduced the Shark Management Strategy; a multidisciplinary approach using new and existing bather protection tools such as Shark-Management-in-Real-Time (SMART) drumlines, acoustic and satellite tagging technology, and drone surveillance [8,13,21,22,23]. As part of this strategy, a large tagging program was initiated to fill knowledge gaps on the movement behaviour and ecology of white sharks in eastern-Australian waters. Results of this help develop alternative and non-lethal methods to mitigate human-shark interactions while also minimising harm caused to sharks and other animals [22,24]. 

As of May 2022, 715 white sharks have been tagged by NSW DPI, where ~90% of tagged individuals were under 300 cm TL (pers com—Paul Butcher—NSW DPI). While previous studies have shown the ontogenetic behaviour of immature eastern Australasian white sharks to be primarily shelf-orientated [7,8,12,13,15,16,25], the scarcity of subadult and adult white sharks encountered in the EAP has led to knowledge gaps in the movement behaviour and habitat use of these age classes. Yet, to accurately inform management programs aimed at ensuring bather protection whilst simultaneously achieving conservation efforts, it is essential to understand movement and habitat use of white sharks across their entire life history 

This study hence aims to characterise broad-scale movements of subadult and adult white sharks, and to examine habitat use between coastal and pelagic environments in eastern Australian waters. The objectives of this study are to (i) quantify horizontal and vertical movement and (ii) identify the influence of depth and temperature on the use of these habitats using pop-up satellite archival transmitting (PSATs) and acoustic tags. Results from the present study provide detailed information on the movement of larger white sharks and are essential for developing bather protecting programs which aim to minimise the risk of shark bites.

## 2. Materials and Methods

### 2.1. Shark Capture and Fishing Gear

Targeted fishing for white sharks occurred between February and September 2021 using Shark-Management-Alert-in-Real-Time (SMART) drumlines [26]. Sharks were caught at Lennox Head (28°48′ S 153°36′ E), Ballina (28°50′ S 153°33′ E) (Figure 1a) and Evans Head (29°10′ S 153°43′ E) (Figure 1b), New South Wales, Australia. SMART drumlines were deployed during daylight hours (between 08:00 and 18:00 AEST), 500 m offshore and in ~10 m of water on sandy substrate following protocols outlined in Tate et al. [24]. All sharks were attended to within 30 min.

### 2.2. Tagging Method

All sharks remained submerged throughout the tagging process with the head positioned forward and body secured parallel to the side of the vessel with a cross-pectoral fin and tail rope [22] (Figure 1c). Sex was determined by the presence (male) or absence (female) of claspers, and total length (TL), was measured to the nearest cm from the point of the shark’s snout to the end of the upper lobe of the caudal fin using a measuring tape. Maturity status assigned to tagged individuals in this study are according to the life history descriptions outlined by Bruce and Bradford [15]. Additionally, the date, time of boat arrival (hh:mm), time the animal was secured at boat (hh:mm), time of animal release (hh:mm), GPS coordinates of capture and release location (decimal degrees), water depth (m), sea surface temperature (°C) and distance offshore (m) were recorded.

### 2.3. Tag Details and Programming

White sharks were tagged with MiniPAT pop-up satellite archival transmitting tags (PSATs; Wildlife Computers, Redmond, WA, USA) that recorded ambient light-level, water depth (pressure range 0–2000 m; accuracy ± 1% of reading; resolution 0.5 m) and temperature (range −40 to 60 °C; accuracy ± 0.1 °C; resolution 0.05 °C) (Figure 2a). Tags were programmed to record and archive a time-series of the aforementioned parameters every 3s with a sample interval of 5 min. Recovery of the tag provided access to the archived 3s data recordings, as well as the 5 min summarised data for analysis. Summary data consisted of depth-temperature profiles, time-at-depth (TAD; pre-programmed bins), time-at-temperature (TAT) and light-level curves. MiniPAT tags had a pre-programed deployment of 120 d (*n* = 3), after which they released from the animal via a corrodible pin, allowing the tags to float to the water’s surface and commence transmission of archived data to the ARGOS network.

MiniPATs (12 cm length, volume 60 cm^3^) were fitted with a Domeier dart head [27] and inserted approximately 10 cm into the musculature at the base of the first dorsal fin using a handheld tagging pole (Figure 2b). Tags were inserted on a 45° angle towards the shark’s head to ensure tags remained in a trailing position on the body. 

To obtain higher resolution location data and form part of a broader tracking program implemented by the NSW DPI, all sharks were externally tagged with a V16-6L Vemco acoustic transmitter (Innovasea, Bedford, NS, Canada) (Figure 2b). These tags have transmission intervals of 40–80 s, and provide long term data (battery life up to 10 years) on horizontal movement. Acoustic transmission is retrieved through acoustic receivers located in Australian waters which form part of the Integrated Marine Observing System’s national network. The transmitters (encased in epoxy, 11.5 cm in length), were fitted with a Domeier dart head [27] and attached as described above. Additionally, all sharks were fitted with a uniquely numbered conventional tag (spaghetti tag, Hallprint Pty Ltd., SA, Australia), used to assist in identification upon recapture (Figure 2b). Conventional tags were inserted approximately 5 cm into the musculature at the base of the first dorsal fin using a hollow cannular. 

### 2.4. Track Reconstruction and Data Analysis

Archived data from all three MiniPAT tags were prepared in the manufactures data portal and light-based geolocation analysis was performed using the manufacturers proprietary Hidden Markov Model (HMM) [28] to estimate daily positions. This approach uses light-level, sea surface temperature and depth readings collected by the MiniPAT tags to generate time-discreet and gridded probability distributions to estimate the most likely positions throughout tag deployment. As the HMM failed to converge, tracks were thinned 1 d at a time until model convergence was achieved, and to avoid clustering. HMM outputs achieved similar scores using speeds of 1 to 5 m s^−1^. Therefore, the speed filter was set to 5 m s^−1^ to avoid spatial constraint of model likelihoods. The inclusion of known locations (acoustic detections) was also used to interpolate missing location data. 

Track lengths were generated from Google Earth after plotting known daily positions from the HMM outputs. Track length divided by days at liberty provided an estimate of average movement rate per day. Depth and temperature data recorded by MiniPAT tags were aggregated into bins and analysed using histogram analysis. Time-at-depth (TAD) data were aggregated into 50 m bins (range 0 to 650 m), and time-at-temperature (TAT) data were aggregated into bins starting at 0 to 9 °C, with subsequent bins increasing by 2 °C (range 0 to 25 °C). Sharks W1 and W3 (Table 1) were removed for TAT analysis due to lack of temperature data from unsuccessful tag retrieval. To investigate diel differences of vertical habitat use, depth and temperature records were further aggregated into periods of day (08:00 to 18:00 h) and night (18:00 to 08:00 h). Welch’s two sample t-tests were performed to test for significant diel differences in mean depth and temperature. Additionally, time-series of depth were plotted over the entire length of deployment for each shark to further examine vertical behaviour. All analyses were performed in R Statistical Software, version 4.1.0 [29]. Archived depth and temperature data were analysed with the R package ‘RchivalTag’ [30], and the R package ‘crawl’ [31] was used to interpolate location data. 

## 3. Results

### 3.1. Tag Success and Deployment Summarys

Three (two male and one female) white sharks, ranging from 340 to 388 cm TL (mean 369 cm TL, SD ± 33.94) were tagged with MiniPAT pop-up satellite archival transmitting tags (PSATs), and acoustic transmitters (Table 1). One male (W1), was classified as adult (>360 cm TL for male), measuring 379 cm TL. The other individuals, one female (W2; 388 cm) and one male (W3; 340 cm), were classified as subadults, based on >300–360 cm TL for male; and >300–480 cm TL for female.

All three MiniPAT tags remained attached for the entire 120 d deployment. Positional information and depth data were successfully obtained from all three tags, while temperature data was provided from one shark, W2, due to the archival data being accessed after successful tag retrieval. W1 and W3’s tag popped-up offshore, ~200 km east of the New South Wales and Victorian border (37°43′ S, 152°18′ E), and ~33 km from the coast of Mount William National Park, northeast Tasmania (41°02′ S, 148°39′ E), respectively, resulting in unsuccessful tag retrieval and subsequent loss of temperature data. W2’s archived water temperature spanned a 16.35 °C range from 7.70 to 24.05 °C (mean 16.79 °C, SE ± 0.002) (Table 2). Individual movements ranged from 6243 to 10,420 km (mean 8761 km, SD± 2217.03) (Table 1), with a maximum dive depth of 644.5 m (Table 2).

### 3.2. Horizontal Movement and Behaviour

MiniPAT tagged white sharks occupied tropical and temperate waters of the Coral and Tasman Seas and Bass Strait. Latitudinal movements ranged from the Capricorn and Bunker Group, Queensland (23° S) to northeast Tasmania (41° S), with an estimated longitudinal range from the coast to 155° E (Figure 3). Average movement rate per day of W1, W2 and W3 were 86.83, 52.02 and 80.16 km day^−1^, respectively, with W1 travelling the greatest distance of an estimated 10,420 km (Table 1). Post-release, all three sharks moved offshore (>20 km) in an easterly direction, before W1 and W2 moved north into Queensland waters, and W3 moved in a south-easterly direction remaining in New South Wales waters. W1 traversed offshore waters (~80 to 280 km) between Agnes Water and Yeppoon, until reaching its northern most point in the Capricorn and Bunker Group, Queensland (23°7′ S, 151°55′ E). W2 reached its northern most point ~74 km east of Teewah Beach (26°10′ S, 153°49′ E), before traversing offshore waters (~74 to 120 km) between Teewah Beach and Sunshine Coast, Queensland. During the tracking period, W3 did not move north past its tagging location, hence, Evans Head (29°10′ S, 153°43′ E) is the northern most point recorded. 

A pattern of seasonal movement was evident during the Austral winter (June to August). Both W1 and W2 made direct southerly movements during July, dispersing from warmer southern Queensland waters, occupying southern New South Wales and Victorian waters by the end of August. During its southerly migration, W1 reached its eastern most point of 155° E, ~315 km offshore from Sydney, New South Wales, before moving closer to the shelf edge and continuing south. W2 entered shelf waters ~3 km from North Stradbroke Island, New South Wales, ~160 km north of its tagging location; a second time reaching its southern most point in eastern Bass Strait near Flinders Island, Tasmania (39°22′ S, 148°4′ E), and a third time, ~4 km from the coast of Bermagui, New South Wales, where it’s tag pop-up location was recorded (36°28′ S, 150°7′ E). W1 and W2 continued to traverse offshore waters in the Tasman Sea during the Austral Spring (September to November), between southern New South Wales, eastern Victoria and northeast Tasmania, until tag release in September and October, respectively. This strong seasonal pattern in horizontal movement is less evident with W3. Unlike W1 and W2, which bypassed most of the New South Wales coast on their migration south, W3 continuously moved up and down the coast of New South Wales, reaching Victorian waters by late October, before moving north again into New South Wales. W3 reached its maximum longitudinal range of 154° E, ~133 km from the coast of Port Macquarie, New South Wales, and made numerous shelf occurrences coming within <5 km of the coast. By early December, W3 was back in Victoria and traversed shelf and shelf-edge waters between Victoria and Tasmania until tag pop-up ~33 km from the coast of Mount William National Park, northeast Tasmania (41°02′ S, 148°39′ E).

W1 and W2 swam similar routes, primarily utilising pelagic and shelf-edge/slope waters and were rarely shelf-orientated. Although W3 utilised the same north–south corridor as W1 and W2, differences of horizontal movement behaviour were visible. Post-release, W3 moved south instead of north, in addition the shark regularly entered and utilised shelf waters and made frequent directional changes over smaller spatial scales. W1 obtained the greatest latitudinal and longitudinal range, exhibiting a broader spatial scale than W2 and W3. 

### 3.3. Depth, Temperature and Vertical Diel Habitat Use

Archived MiniPAT depth data of revealed that white sharks generally occupied the epipelagic zone (0 to 200 m) during the day (W1 ~80%, W2 ~84% and W3 ~97%) and at night (W1 ~93%, W2 ~92% and W3 ~100%) (Figure 4). Particularly, the upper 50 m of the water column was favoured (day: W1 ~38%, W2 ~48% and W3 ~74%; night: W1 ~48%, W2 ~57% and W3 ~83%). Movements into the mesopelagic zone (200 to 1000 m) were linked with deep diving behaviour. Minimum and maximum depths ranged from 0 to 644.50 m. W1 and W2 dove to maximum depths with only 4.5 m difference at 640.00 m and 644.50 m, and logged mean depths (±SE) of 84.77 ± 0.60 m and 79.20 ± 0.60 m, respectively (Table 2). W3 dove to a maximum depth of 604.50 m, and logged a mean depth (±SE) of 32.59 ± 0.38 m, notably shallower than W1 and W2. Below the epipelagic zone, W1 appeared to favour depths between 200 to 350 m, occupying these waters ~26% of the time during the day. In comparison, W2 occupied these depths only ~8% of the time. However, W2 utilised depths between 350 to 500 m ~10% of the time during the day, compared to W1 (~1%). W3 rarely entered the mesopelagic zone, spending only ~3% of the time in depths greater than 200 m during the day (Figure 4c). 

TAD and TAT data recorded by W2, reveal a clear relationship between water temperature and depth, recording higher temperatures in surface waters, and lower temperatures with increasing depths (Figure 4b and Figure 5). Welch’s two sample t-test revealed W2 spent significantly more time in lower temperatures during daylight hours and higher temperatures at night with mean temperatures (±SE) 16.23 ± 0.002 °C and 17.27 ± 0.002 °C, respectively (t = −289.5, df = 3425332, *p* ≤ 0.05) (Figure 5). Water temperatures occupied by W2 ranged from 7.70 to 24.05 °C (mean 16.79 ± 0.002 °C), with a clear preference for temperatures between 14 to 19 °C, spending ~65% of the time in this range during the day and at night. Outside of this range, W2 favoured temperatures between 22 to 23 °C, spending ~14% and ~18% of the time in this range during the day and at night, respectively. Temperatures below <12 °C were rarely occupied, with a minimum temperature of 7.70 °C recorded during W2’s deepest dive of 644.50 m. 

Mean depths (±SE) occupied by W1, W2 and W3 during daylight were 105.45 ± 1.06 m, 95.28 ± 1.00 m and 39.09 ± 0.61 m; and, 68.20 ± 0.62 m, 65.35 ± 0.68 m and 24.66 ± 0.41 m at night, respectively. All sharks’ daylight values were found to be significantly different from night values (W1: t = 30.394, df = 18584, *p* < 0.05; W2: t = 24.738, df = 28880, *p* < 0.05; and W3 t = 19.501, df = 21556, *p* < 0.05), indicating all three sharks preferred deeper water during daylight hours and shallower waters at night. This further supports W2’s TAT data, and the relationship between temperature and depth. In particular, cooler temperatures were associated with diving behaviour and greater depths. Furthermore, all three sharks took their deepest dives during daylight (Figure 6). 

Depth profiles revealed similar diving behaviour with W1 and W2. Both sharks performed the majority of their deep diving in the first half of their 120 d tag deployment, frequently entering the mesopelagic zone (Figure 7a,b). A change in vertical behaviour was evident approximately halfway through the deployment, with W1 and W2 primarily occupying the epipelagic zone once they reached the mid-south New South Wales coast. This change in vertical behaviour continued for the remainder of tag deployment, as sharks traversed offshore waters between southern New South Wales, eastern Victoria and northeast Tasmania. In contrast, W3’s depth profile revealed no pattern in deep diving behaviour with very few deep dives into the mesopelagic zone (Figure 7c). The few deep dives W3 did take were in pelagic and shelf-edge waters in New South Wales and one in Victorian waters ~223 km east of the coast of Mallacoota. Furthermore, depth profiles reveal W2 utilised deeper depths more frequently than W1 and W3, supporting TAD histograms.

## 4. Discussion

This study describes movement behaviour and habitat use of subadult and adult white sharks throughout tropical and temperate waters of eastern Australia. It provides details of the first adult individual tagged with multiple sensors in the New South Wales Department of Primary Industries (NSW DPI) shark tagging and tracking program, expanding our knowledge of the movement behaviour and habitat use of EAP white sharks across all life stages. Although considered a temperate species, findings support previous studies revealing white sharks regularly occupy sub-tropical and tropical waters [1,8,12,13]. Horizontal movement of all three sharks demonstrated a north–south seasonal pattern along the eastern seaboard of Australia, and the highly mobile nature of the species, covering long distances over the short temporal period of the study. Broad-scale movement varied among the three conspecifics depending on season, but overall movements of white sharks revealed a similar latitudinal and longitudinal extent with a preference for offshore habitats along the relic continental shelf. All three individuals favoured depths within the upper 50 m of the water column and W2 favoured temperatures between 14 to 19 °C. Although the described preferences are within a limited range, tolerances for both depth and temperature were broad. Additionally, diel patterns of vertical behaviour, featuring deeper depths occupied during the day and shallower depths at night, were consistent for all three individuals. 

### 4.1. Horizontal Movement

#### 4.1.1. Longitudinal Movement

Post release, all three sharks moved east from the coast (>20 km), before heading in a northerly or southerly direction following the Australian coastline. This offshore movement post capture ‘fright and flight’ response after tagging has been recorded in white sharks caught from the same area [34] and other species that occupy eastern Australian waters, including Tiger shark, *Galeocerdo curvier* [35], Sandbar, *Carcharhinus plumbeus* and Dusky, *C. obscurus*, sharks [36]. Tagged white sharks have also been documented to have an inshore phase immediately after release off the coast of central California [37]. Two recent studies investigating the physiological stress response of juvenile and subadult white sharks of the EAP to SMART drumline capture, observed no significant change of physiological variables [22,38]. This indicates that a typical response to capture and tagging is a relatively benign process, and the immediate offshore movement presented here can be considered normal behaviour. 

Corroborating previous research of subadult and adult white sharks, tracked individuals in this study also spent most of the time in offshore and pelagic waters [4,7,37,39]. W1 recorded the greatest longitudinal point of 155° E, with W2 and W3 recording a maximum of 154° E. Although far offshore (~315 km), this is not the greatest longitudinal extent reached of EAP white shark. Juveniles have been tracked performing long-distance migrations to sub-Antarctic waters [13,40], and crossing oceanic basins is not uncommon of the species. For example, a 380 cm TL female crossed the Indian Ocean from Gansbaai, South Africa to Exmouth Gulf, Western Australia [14]. W3 exploited the greatest longitudinal range of all three individuals, utilising the continental shelf and slope, as well as the offshore and pelagic habitats primarily occupied by W1 and W2. It is likely the longitudinal movements of W3 were associated with prey searching. The East Australian Current (EAC) produces nutrient rich upwellings that encroach on continental shelf and slope waters increasing primary productivity and foraging opportunities to these areas [41,42,43]. Similar longitudinal movement has been documented for tiger sharks in eastern Australia and were attributed to foraging and resource availability [35]. 

Overall, longitudinal movements of the three individuals support the 2-population model proposed by Blower et al. [9], as no sharks made east–west movements through Bass Strait. However, the small temporal scale and sample size of this study should be considered as spatial division of juvenile EAP white sharks from the southern-western region were more accurately represented with an increased sample size (*n* = 87) [13]. Consequently, the interaction of subadult and adult EAP within the SWP may not be clearly represented until greater sample numbers are reached.

#### 4.1.2. Latitudinal Movement

Offshore latitudinal movements of white sharks were characterised by highly directional swimming and long distances between locations, indicating travelling behaviour [42]. This behaviour was recognised as seasonal movement, as all three individuals utilised a north–south corridor proposed by Bruce et al. [1], moving from southern Queensland and northern NSW waters to Victorian and Tasmanian waters. W2 remained over the continental shelf edge and slope, only utilising adjacent waters >50 km once entering the southern NSW, eastern Victoria and northeast Tasmanian waters. As such, bathymetric features, like the shelf edge and slope, and their geophysical gradients may assist in long-distance movement [44]. For example, the highly directional swimming to and from sea mounts performed by Scalloped Hammerhead sharks (*Sphyrna lewini*) in the Gulf of California were attributed to the sharks’ ability to use the magnetic field of these features for navigation [45].

Seasonal movement has also been described in white shark populations in the north Atlantic [4], and northeast Pacific [39]. Individuals >300 cm were reported displaying a less defined pattern of seasonal movement in offshore and pelagic habitats [4]. In contrast, the present study found the two larger individuals, W1 (379 cm TL) and W2 (388 cm TL) to display this pattern more strongly than W3 (340 cm TL). A likely explanation for this difference in seasonal movement behaviour could be due to the difference of time of year and season of tagging. W1 and W2 were both tagged in Austral Autumn and Winter months (May and June), while W3 was tagged in the Austral Spring (September) when W1 and W2 were already occupying Victorian and Tasmanian waters. Irrespective, the present study reveals subadult and mature white sharks of the EAP continue to display a seasonal migration pattern while in offshore habitats. Furthermore, W1 was detected on an acoustic receiver ~6.5 km offshore from Moonee Beach, New South Wales (30°12′ S, 153°13′ E), only 6 weeks after tag pop-up and ~160 km south of tagging location, further supporting seasonal latitudinal movement of white sharks. 

Overall, horizontal movements of white sharks display the broad-scale longitudinal and latitudinal range of the EAP. W1 and W3 achieved similar average movement rates per day of ~86.83 and ~80.16 km/day, followed by W2 at ~52.02 km day^−1^. Comparable movement rates have been previously recorded of 71 km day^−1^ [37] and 74 km day^−1^ [39], with some studies reporting speeds up to 112 km day^−1^ [14] and 119 km/day [2]. Furthermore, the large distances achieved by W1, W2 and W3 of ~10,420 km, ~6243 km and ~9620 km, respectively, in the 120 d study period are not uncommon for this species. A similar distance of ~11,100 km was recorded by a subadult female (380 cm TL) in 99 days [14]. As such, this study highlights the transient potential and need for multijurisdictional conservation and management programs, as the broad-scale horizontal movement of white sharks encompassed each state along the eastern seaboard of Australia.

### 4.2. Resident Behavior

Direct movements of all three individuals alternated with localised movements over a smaller spatial scale, characterised by shorter distances between locations indicating areas of residency [43]. This is particularly evident as W1 and W2 occupied southern Queensland waters in the months of June and beginning of July. During the period of residency, W1 traversed offshore shelf and shelf edge waters adjacent to the Capricorn and Bunker Group, and came within close proximity to the reef network which make up the southern extent of the Great Barrier Reef (~2000 km stretch of reef). This area was documented as an area of residency for an adult male (360 cm TL) from the SWP also during the Austral Winter [1], and has been recorded as a peak area of residency for juvenile white sharks of the EAP [12,19]. W2 traversed waters over the shelf edge and slope ~250 km south of W1. The utilisation of highly productive slope waters is likely attributed to prey availability and foraging. The separation of the residential areas of W1 and W2 in southern Queensland may be a result of sexual segregation, as it has been previously recorded for male and female white sharks to have different distributions within a shared offshore foraging area in the northeast Pacific [39]. 

An additional area of interest was identified for W1 and W2 located in the Tasman Sea during the Austral Winter and Spring (late July to beginning of October). The large area encompassed offshore waters of southern New South Wales, eastern Victoria and northeast Tasmania. Such a large area may not be recognised as residency, as the distances between turning angles are greater than what is recognised as area restricted movement [43]. However, it is evident W1 and W2 seasonally occupied this area following their direct migration south. This area appears to attract EAP white sharks of all life stages, as juvenile individuals have also been recorded occupying or traversing through the area similarly to W3 [12,13]. Below the point of separation of the EAC from the coast, is a dynamic moving eddy field referred to as the ‘Eddy Avenue’ [42]. Properties of eddies include increased chlorophyll-a concentrations, enriched with plankton and larval fish that subsequently play an important role in primary productivity [41]. As such, this location may be an important offshore foraging area for EAP white sharks. The months of residency for W1 and W2 in the area occurred earlier than peak months recorded for juveniles and W3 (November-February) [12,13]. This may indicate that this area does not only provide opportunities for foraging, but it may also be a focal offshore area for reproduction. The reproductive strategies and dynamics of white sharks remain poorly understood [18]. Determining whether this area plays a significant role in reproduction of the EAP should be prioritised in future studies. 

The locations utilised by W3 in northern New South Wales waters during the Austral Spring, and eastern Victoria/northeast Tasmania during the Austral Summer, corroborate juvenile nursery areas and seasons of occupancy [8,12,13,15]. Sea mullet (*Mugil cephalus*) and Australian salmon (*Arripis trutta*) are abundant in coastal waters of New South Wales most of the year, and the residential timing of W3 in eastern Victoria corresponds with the snapper (*Chrysophrys auratus*) breeding season (October–April). All three teleost species are known prey items for juvenile white sharks [46]. White sharks experience an ontogenetic shift of target prey items, moving from a diet primarily consisting of teleost fin fish, to include marine mammals once reaching lengths > 300 cm TL [46]. Although W3 is mature enough to prey upon marine mammals, it is likely that the individual took advantage of the available prey resource in the area. Furthermore, the offshore areas utilised by all three individuals are not in the vicinity of pinniped colonies, which suggests, teleosts may remain an important dietary item for eastern Australasian white sharks across all life stages, and can indirectly influence seasonal movement and residency. Future research using environmental DNA from cloaca swabs to understand the diet of sharks would provide a unique opportunity to formalise these movement patterns further [47]. 

### 4.3. Depth and Temperature Preference

Throughout the tracking period, white sharks primarily occupied the epipelagic zone (0–200 m), and displayed a clear preference for the upper 50 m of the water column during both daylight and night time periods. This is consistent with previous research and further supports a preference for this depth across the species’ entire life history and circum-global range [1,2,4,7,48]. W1 and W2’s overall mean depths of 84.77 m and 79.20 m, respectively, were notably deeper than W3’s mean deployment depth of 32.59 m. The similarity of W1 and W2’s behaviour in comparison to W3 could be due to tagging occurring at a different time of year and season. This highlights the need for sampling to occur throughout the year and across different seasons to obtain a better understanding of the differences of behaviour represented here. 

W2 displayed a clear preference for a temperature range between 14 to 19 °C. This is consistent with previous findings of the EAP [8,24,43,48], and of white shark populations in South Africa [49] and the north Atlantic [4]. White sharks have a physiological adaptation that enables core body temperatures to remain higher than the surrounding sea water [50]. This adaptation allows the species to travel to high latitudes and enter cold, nutrient rich waters in search of prey. However, the consistency of the favoured temperatures shown here, and in previous studies, indicate an optimal thermal niche of white sharks. Thus, temperature may be considered a driver for habitat preference outside of primary productivity and prey items. Indeed, water temperature has been identified as a driver for movement in other shark species including: *Rhincodon typus* [51], *G. cuvier* [35] and *Isurus oxyrinchus* [52]. Furthermore, the preferred temperature range experienced by W2 may also serve as an indicator for presence of white sharks in eastern Australian waters. For example, the probability of white shark sightings in South Africa has been shown to increase as sea surface temperatures (SST) exceeded 14 °C [48].

Outside of the preferred thermal range, W2 spent a considerable amount of time in temperatures 22 to 23 °C. These temperatures were recorded in southern QLD and northern NSW waters, and are the average water temperatures experienced in these regions during winter months [53]. Lower temperatures were recorded as W2 migrated south, reaching higher latitudes and when deep diving. As a result, W2 experienced a broad range of temperatures (7.70 to 24.05 °C), supporting previously documented temperature ranges and thermal tolerances of the species [7,8,34,37,43,48,49,54]. Previous studies have speculated sex biases of temperature preference in white sharks [7,39]. Due to the limitation of physical MiniPAT tag recovery, the comparison of temperature preference between sexes is beyond the scope of this study. A difference in temperature range and preference may have been present between conspecifics, considering the differences of mean depths and time of year and season of tagging. Further research should continue to quantify this variable and investigate sex-biases of subadult and adult individuals of the EAP to elucidate this supposition. Understanding the temperature preference of white sharks across all life stages and sexes will increase our predictive capabilities of occurrence.

### 4.4. Vertical Habitat Use and Diving Behaviour

All three individuals demonstrated diel patterns of vertical habitat use, featuring shallower depths occupied at night and deeper depths during the day. This pattern has been previously reported in white sharks [48,54], yet variation among individuals does exist. Diel patterns have been attributed to foraging behaviour following diel vertical patterns of nekton [43], with intraspecific and interspecific variation likely a result of different foraging strategies and target prey species [48,54,55]. 

W1 and W2 displayed plasticity in their diving behaviour. During the first half of tag deployment, both individuals regularly dove into the mesopelagic zone. This included the identified areas of residency in southern Queensland and mid to northern New South Wales waters during their southerly migration. Subsequently, as both sharks reached offshore waters east of Sydney, New South Wales, they primarily occupied the epipelagic zone with few random deep dives below 200 m until tag pop-up. Similarly to vertical habitat use, previous studies investigating vertical movement of white sharks and other shark species have attributed diving behaviour to foraging and navigation during migration [35,36,44,45,48,56]. It is also hypothesised diving behaviour assists in thermoregulation, as most shark species are ectothermic and require the warming effects of surface waters after diving into deep, cold waters [51]. However, as white sharks are one of the few endothermic shark species with the ability to regulate their body temperature [57], it is likely the diving behaviour presented here is a result of the former. In contrast, W3 remained in the epipelagic zone with very few dives into the mesopelagic zone for the entire 120 d deployment period. This individual may have already altered diving behaviour based on the influence of extrinsic biotic and abiotic factors that may have occurred later in the year when W3 was tagged. Results suggest, similarly to horizontal movement, white sharks have the ability to rapidly change movement and foraging behaviours to exploit resources available in shifting spatiotemporal extents, as has been reported for other endothermic lamnid species [1,34,56]. Furthermore, vertical plasticity represented here, indicates conservation and management programs of white sharks may require different and alternating protocols depending on different times and seasons of the year. 

## 5. Conclusions

This study highlights the highly dynamic movement behaviour and complex habitat use of subadult and adult white sharks in eastern Australian waters. Variation in horizontal and vertical habitat use between conspecifics were likely attributed to foraging and resource availability, water temperature, season and navigation. To gain a greater understanding of the movement behaviour of these life stages, it is prudent that future studies increase the sample size and quantify movement patterns throughout the year. Noting that the types of tags used in current experiment have a 180 day maximum battery life. However, with ongoing sampling and increased sample size we can elucidate behavioural differences of seasonal latitudinal and longitudinal movement, and diving patterns in these larger immature and mature life stages. This will provide insight into the behavioural variation between conspecifics discussed here, particularly, W3’s behaviour in comparison to W1 and W2. Furthermore, whilst temperature is considered a driver for horizontal movement, it is likely movement and habitat use are driven by the synergistic effects of compounding extrinsic and intrinsic factors rather than one or few factors alone. To improve predictability models of subadults and adults of the EAP, future research should incorporate additional extrinsic environmental parameters that have been investigated in juvenile white sharks of the EAP [8]. 

Globally, unprovoked shark bites are steadily increasing [58,59]. Although the individuals in the present study favoured offshore habitats, all three sharks entered nearshore coastal waters, increasing the likelihood of human-shark interactions in eastern Australian waters. The documentation of movement and behavioural data has improved the predictive capabilities of white shark occurrence across all life stages. As such, findings have expanded our knowledge of the EAP and can be used as a baseline for subadult and adult individuals. This critically important information can be used to inform sound white shark conservation and management practices.

## Figures and Tables

**Figure 1 biology-11-01443-f001:**
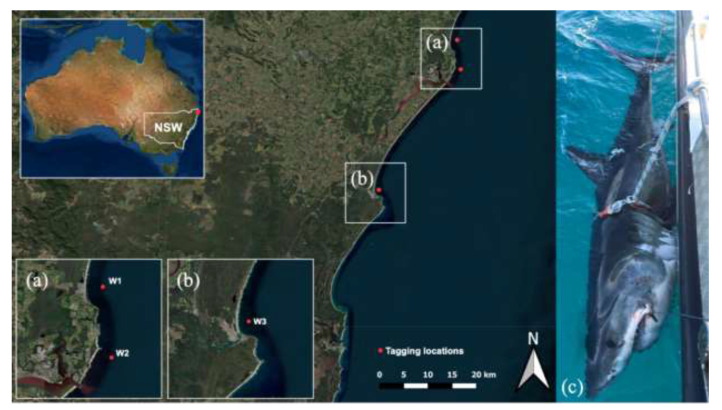
Location of tagging sites (red dots) on the northern New South Wales coast of eastern Australia. (**a**) Lennox Head, tagging site of white shark 1 (W1) and Ballina, tagging site of white shark 2 (W2), (**b**) Evans Heads, tagging site of white shark 3 (W3), and (**c**) W1 secured to the side of the vessel during tagging procedure (image provided by the New South Wales Department of Primary Industries). Map generated in QGIS.

**Figure 2 biology-11-01443-f002:**
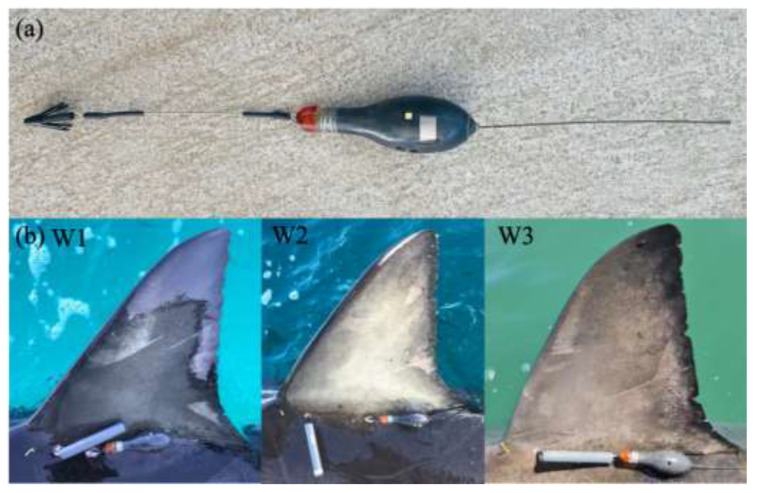
(**a**) MiniPAT-348 Pop-up Satellite Archival Transmitting Tag and Domeier dart head used to anchor the MiniPAT tags and acoustic transmitters into the sharks musculature, and (**b**) dorsal fins of the three white sharks used in the present study. Images taken after completing tagging procedure, with MiniPAT tags, acoustic transmitters and conventional tags visible (images provided by the New South Wales Department of Primary Industries).

**Figure 3 biology-11-01443-f003:**
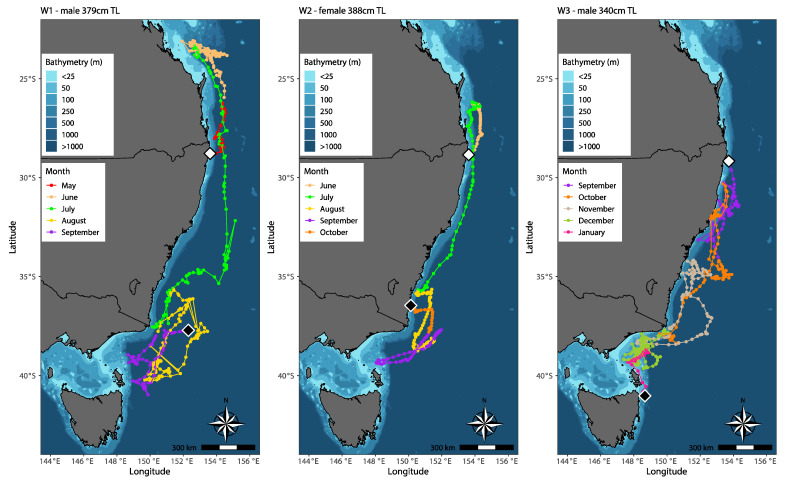
Track reconstruction of three white sharks (*C. carcharias*) tagged off the northern New South Wales coast of eastern Australia. Estimated track positions are colour-coded by month. Tagging locations indicated by white diamonds, and MiniPAT tag pop-up locations indicated by black diamonds. Maps were generated in R Statistical Software, version 4.1.0 [29], using R packages ‘marmap’ and ‘ggplot2’ [32,33].

**Figure 4 biology-11-01443-f004:**
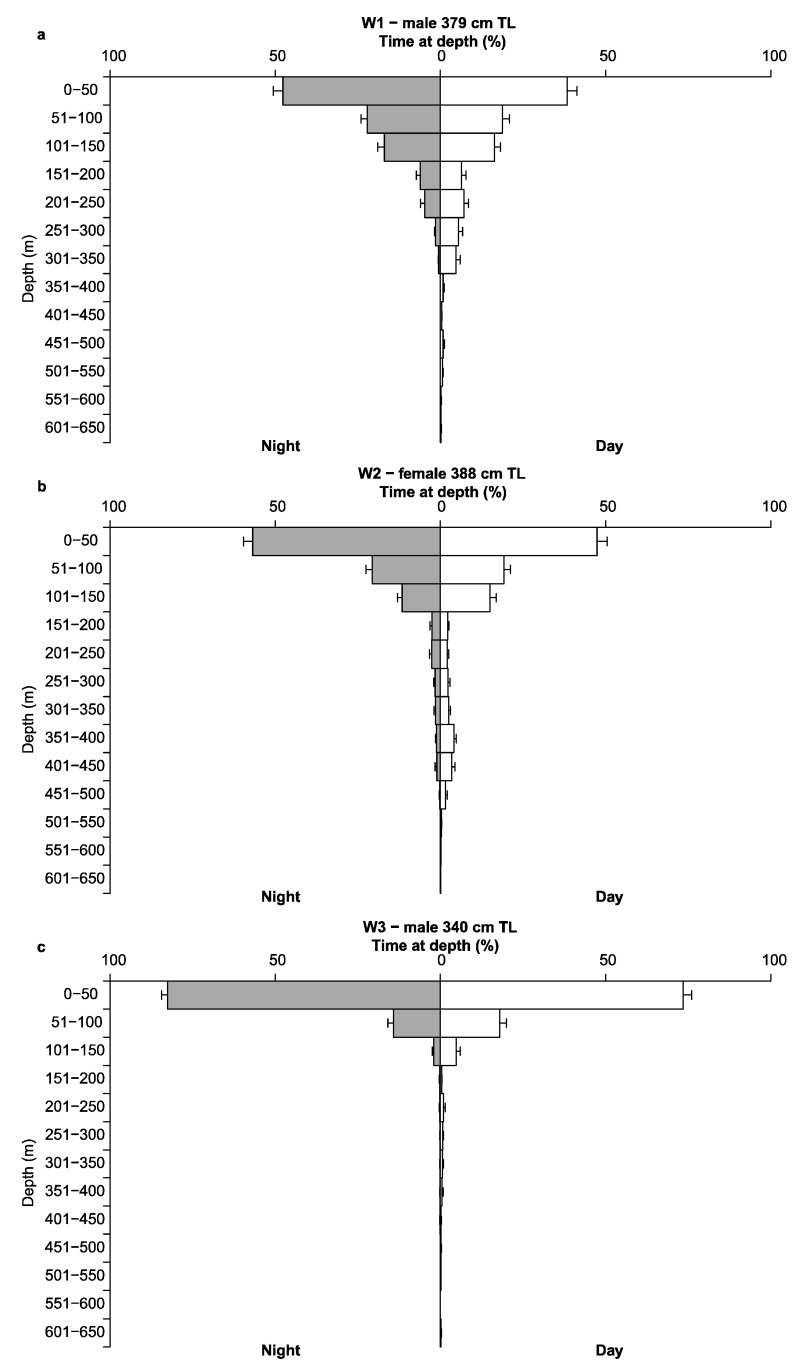
Time spent (%±SE) at depth (m) during the day (white bars) and at night (grey bars) for three white sharks (*C. carcharias*) (**a**) W1, (**b**) W2 and (**c**) W3.

**Figure 5 biology-11-01443-f005:**
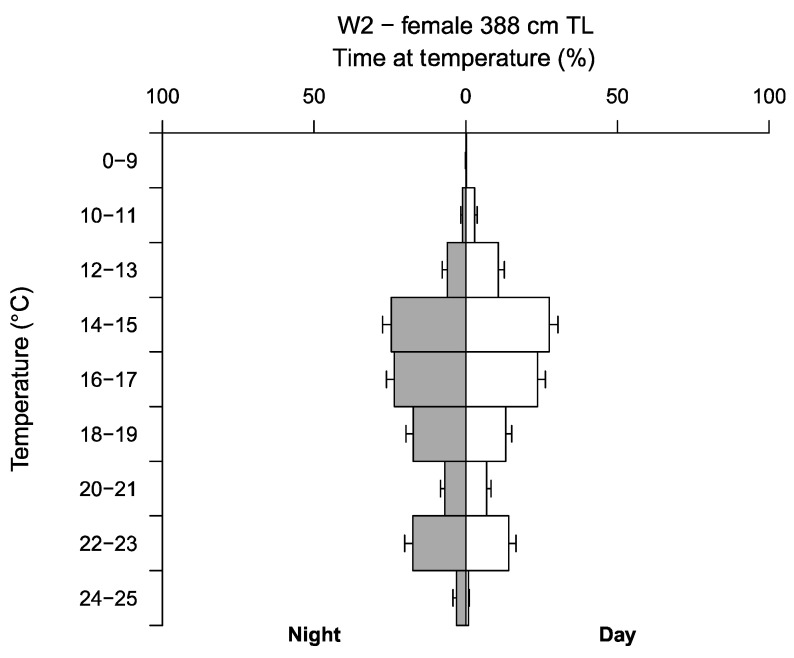
Time spent (%±SE) at temperature (°C) during the day (white bars) and night (grey bars) for one white shark (*C. carcharias*) individual (W2).

**Figure 6 biology-11-01443-f006:**
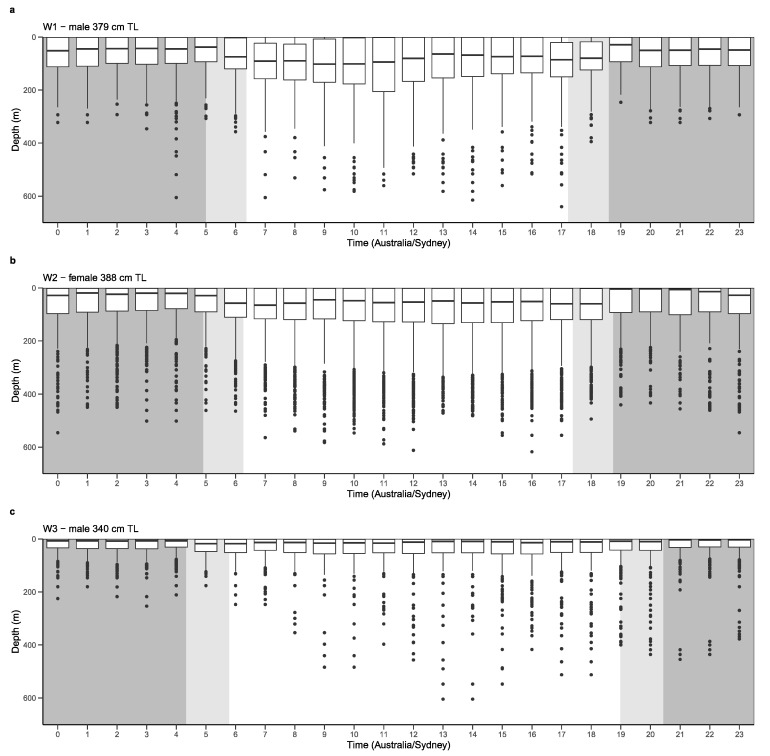
Mean depths (m) of three white sharks (*C. carcharias*) (**a**) W1, (**b**) W2 and (**c**) W3 plotted over a 24-h period (0–23 h AEST). White areas indicate daylight. Areas shaded in grey indicate night-time (darker grey) and twilight (lighter grey) periods.

**Figure 7 biology-11-01443-f007:**
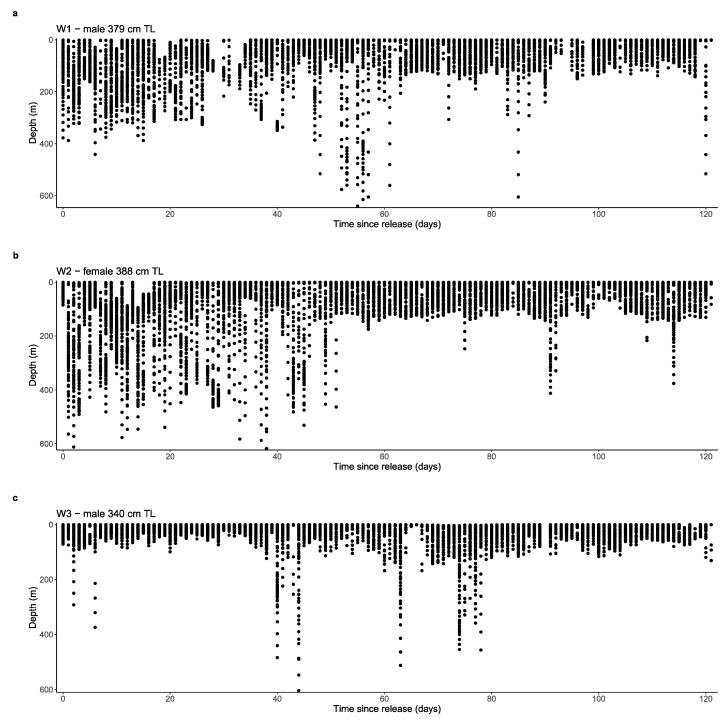
Depth profiles of three white sharks (*C. carcharias*) (**a**) W1, (**b**) W2 and (**c**) W3 across the 120 d MiniPAT tag deployment.

**Table 1 biology-11-01443-t001:** Summary of biological details, tag deployment and tracking details for the three white sharks (*C. carcharias*) tagged off the northern New South Wales coast of eastern Australia. * MiniPAT tag recovered.

ID	Sex	Total Length (cm)	Tagging Date	Tagging Location Lat. (°S), Long. (°E)	Date Tag Detached	Pop-Up Location Lat. (°S), Long. (°E)	Days Tracked	Estimated Track Length (km)	Estimated Average Movement (km day^−1^)
W1	M	379	23 May 2021	28.783, 153.610	20 September 2021	37.721, 152.300	120	10,420	86.83
W2 *	F	388	10 June 2021	28.845, 153.617	8 October 2021	36.468, 150.133	120	6243	52.02
W3	M	340	10 September 2021	29.167, 153.717	9 January 2022	41.026, 148.666	120	9620	80.16

**Table 2 biology-11-01443-t002:** MiniPAT archived data of three white sharks (*C. carcharias*) at depth and temperature. * MiniPAT tag recovered.

Depth (m)			Temperature (°C)	
Shark ID	Min-Max	Mean ± SE	Min-Max	Mean ± SE
W1	0.50–640.00	84.77 ± 0.60	-	-
W2 *	0.00–644.50	79.20 ± 0.60	7.70–24.05	16.79 ± 0.002
W3	0.50–604.50	32.59 ± 0.38	-	-

## Data Availability

Data for this project are maintained by the New South Wales Government and Deakin University, Victoria. The data are available from the authors.

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
