# Peer review of "Preliminary Data about Habitat Use of Subadult and Adult White Sharks (Carcharodon carcharias) in Eastern Australian Waters"

_biology, 2022, doi:10.3390/biology11101443_

Round 1

Reviewer 1 Report

Dear Authors,

your paper is really interesting. You can find minor comments and suggestions in the file attached.

Kind regards

Author Response

Journal of Biology - MDPI

14 September 2022

Dear Cristina (Editor)

I have attached an electronic copy of the revised manuscript (Biology-1926933)  ‘Preliminary data about habitat use of subadult and adult white sharks (Carcharodon carcharias) in eastern Australian waters’.  We have made the revisions and believe it has considerably improved the manuscript.  We have listed our responses to the various comments below and highlighted these in the main document as track changes.

Reviewer #1

1st comment: I recommend that the authors include a paragraph at the end of the discussion or in the conclusion, mentioning this fact and suggesting a larger study with a larger scale tagging and a longer temporal follow-up (over years) to validate the suggested seasonal effects

We agree with the reviewer and have added this into the conclusion.

2nd comment: In view of the large number of abbreviations in the manuscript, I also suggest that the authors provide a table of abbreviations.

This is outside the scope of the journal formatting. However, we have changed many of the abbreviations in the main document but kept those that are used to describe units of measurement (i.e. d – days, m – meters).

3rd comment: Lines 81, 131, 134, 141, etc: please add a space between the measure and the unit. to be reviewed throughout the manuscript

We have made these changes through the manuscript

4th comment: Line 108: presence (male) or absence (female) of claspers

We have made this change

5th comment: Line 166: Please add an abbreviation. proprietary Hidden Markov Model (HMM) [28]

We have made this change

6th comment: Line 172: Please verify all the unit, including superscript numbers

We have made this change

7th comment: Line 198: Please provide the REF

The term ‘REF’ has been removed and the correct sentence structure provided.

8th comment: Line 240 to 261: I agree with the authors on the seasonal pattern of movement of W1 and 2 sharks. As discussed, the W3 shark seems to show less stereotyped movements and move down the Australian coast in a more erratic manner. Could the tagging period be to blame? Tagging does not seem to affect the short-term behavior of these sharks. But what about the long-term behavior. Could the interruption of the "migration" during tagging explain the distinct behavior of this shark?

Thank you for the comment but we argue that at least in white sharks there are no indications of changes in the short or long-term tracking behaviour caused by tagging operations. We have published the following manuscripts:

Gallagher A, Meyer L, Pethybridge H, Huveneers C, and Butcher P 2019. Physiological stress responses of white sharks (Carcharodon carcharias) to short-term capture: amino acids and fatty acids. Endangered Species Research, 40:297-308

Grainger, R.,  Raubenheimer, D., Peddemors, V.M., Butcher, P.A., Machovsky-Capuska, G.E. 2021. Integrating biologging and behavioural state modeling to identify cryptic behaviours and post-capture recovery processes: New insights from a threatened marine apex predator. Frontiers in Marine Science 8: e791185.

Tate, R.D., Cullis, B.R., Smith, S.D.A., Kelaher B.P., Brand, C.P., Gallen, C.R., Mandelman, J.W., Butcher, P.A. 2019. The acute physiological status of white sharks (Carcharodon carcharias) exhibits minimal variation after capture on SMART drumlines . Conservation Physiology 7(1): coz042.

Spaet, J., Patterson, T., Bradford, R., Butcher, P. 2020. Broad scale movement and residency of White sharks in Eastern Australia and New Zealand. Scientific Reports 10 (1): 1-13 

Spaet, J. L., Manica, A., Brand, C. P., Gallen, C., & Butcher, P. A. 2020. Environmental conditions are poor predictors of immature white shark Carcharodon carcharias occurrences on coastal beaches of eastern Australia. Marine Ecology Progress Series, 653: 167-179.

9th comment: Line 317: What TAS means? I suspected Tasmania.

We have removed TAS and replaced it with ‘Tasmania’

10th comment: Line 333: Is figure 5 essential for reading the manuscript? This data is, unfortunately, taken from only one individual. I suggest placing this figure in an appendix.

Figure 5 provides critical information for the reader and results section. We believe that this should remain in the main document.

11th comment: Lines 364 and 554: Looking at figure 7 and excluding W3, it seems that the other two sharks (W1 and 2) drastically decrease their deep dives after two months of tagging (around August). Do the authors have any monitoring of the planktonic fluctuations (bloom or post-bloom events) for these periods where these sharks are found at their specific location? It might be interesting to analyze a potential temporal correlation between these two events. Could fluctuations in surface food supply explain the start and stop of pseudo-seasonal deep dives?

We agree that investigating the relationship between planktonic fluctuations and deep diving behaviour would be interesting and we have looked into this but given the relatively high uncertainty exhibited by light-based geolocations [root mean square errors within ∼80–150 km] and that white sharks can travel great distances, it is unfeasible to rely on geolocation estimates from pop-up satellite archival tags. We hence tried not to overemphasize on exact locations.

12th comment: Line 501: Please italicize species name

The species name has been italicised

13th comment: Line 534: G. curvier

The word ‘cuvier’ has been spelt correctly

Reviewer #2

The reviewer made comments directly on the PDF provided. We have made all changes they listed.

1st comment: Title - Since you recorded data about 3 individuals for the first time, I would modify the title in something like "Preliminary data about....."

We agree with the reviewer and have made this change

2nd comment: Throughout the manuscript add in a space between numbers and their units of measurement

We agree with the reviewer and have fixed those errors highlighted plus others in the mansucript.

3rd comment: Justify the text throughout the manuscript

We agree with the reviewer and have made these changes

4th comment: Line 192 – captitalise the word ‘tag’

We agree with the reviewer and have made this change

5th comment: Line 228 – remove italics on ‘table 1’

We agree with the reviewer and have made this change

6th comment: Line 232 – remove italics on ‘table 2’

We agree with the reviewer and have made this changes

7th comment: Line 433 captilalise the word ‘scalloped hammerhead’

We agree with the reviewer and have made this change

8th comment: Line 478 spell out Oct in full so it reads October

We agree with the reviewer and have made this change

We thank the reviewers and the editor for their comments. We have also made other changes throughout the manuscript and along with the comments believe that the revised version is now acceptable for publication.  Please contact me if you require further changes.

Yours sincerely

Dr Paul Butcher

Principal Research Scientist

New South Wales Fisheries, NSW Department of Primary Industries

Reviewer 2 Report

Dear authors,

With pleasure, I revised the manuscript (MS) “Habitat use of subadult and adult white sharks (Cacharodon cacharias) in eastern Australian waters”. The MS presents information about the biogeographic movements of three white shark specimens along the eastern coast of Australia. The MS is presenting a well-document introduction, a clear material and method section and an extensive results and discussion section. This paper is very interesting, could lead to a better understanding of the ecology of this apex predator in Australian waters, and place the preliminary bases for a greater study on white shark movements along these coasts. Nevertheless, the present study is presenting some gaps, although noted by the authors.

General comments:

The manuscript presents new data on the movements of the great white shark along Australia's east coast. The study using three different tag types corroborates the data already present in the literature regarding vertical and horizontal migrations, habitat preferences, etc. of white sharks. With only three tagged sharks and temperature data available for only one of them, the authors seem aware of the weakness of the results. The increase in the number of replicates is put forward in the discussion in order to validate the "preliminary" results obtained. It would be appropriate to take the conclusion with more caution, particularly by mentioning the need to increase the number of sharks tagged. Similarly, it seems from the manuscript that it is essential to standardize the protocol in terms of tagging timing. Although discussed, it is difficult to interpret the behavior of the W3 shark, the only one tagged in September. I recommend that the authors include a paragraph at the end of the discussion or in the conclusion, mentioning this fact and suggesting a larger study with a larger scale tagging and a longer temporal follow-up (over years) to validate the suggested seasonal effects.

In view of the large number of abbreviations in the manuscript, I also suggest that the authors provide a table of abbreviations.

Specific comments

-       Lines 81, 131, 134, 141, etc: please add a space between the measure and the unit. to be reviewed throughout the manuscript

-       Line 108: presence (male) or absence (female) of claspers

-       Line 166: Please add an abbreviation. proprietary Hidden Markov Model (HMM) [28]

-       Line 172: Please verify all the unit, including superscript numbers

-       Line 198: Please provide the REF

-       Line 240 to 261: I agree with the authors on the seasonal pattern of movement of W1 and 2 sharks. As discussed, the W3 shark seems to show less stereotyped movements and move down the Australian coast in a more erratic manner. Could the tagging period be to blame? Tagging does not seem to affect the short-term behavior of these sharks. But what about the long-term behavior. Could the interruption of the "migration" during tagging explain the distinct behavior of this shark?

-       Line 317: What TAS means? I suspected Tasmania.

-       Line 333: Is figure 5 essential for reading the manuscript? This data is, unfortunately, taken from only one individual. I suggest placing this figure in an appendix.

-       Lines 364 and 554: Looking at figure 7 and excluding W3, it seems that the other two sharks (W1 and 2) drastically decrease their deep dives after two months of tagging (around August). Do the authors have any monitoring of the planktonic fluctuations (bloom or post-bloom events) for these periods where these sharks are found at their specific location? It might be interesting to analyze a potential temporal correlation between these two events. Could fluctuations in surface food supply explain the start and stop of pseudo-seasonal deep dives?

-       Line 501: Please italicize species name

-       Line 534: G. curvier

Author Response

(The authors gave the same response as above.)
